Analysis of deubiquitinase OTUD5 as a biomarker and therapeutic target for cervical cancer by bioinformatic analysis

Bai Mixue
Che Yingying
Lu Kun
Fu Lin fulin@qdu.edu.cn
Institute of Chronic Disease, Qingdao University , Qingdao , Shandong , China
de Azevedo Jr. Walter
Electronic publication date: 2020 Jun 30
Publication date: 2020
Volume: 8
Electronic Location ID: e9146
Received 2019 Dec 30; Accepted 2020 Apr 17
Copyright: ©2020 Bai et al.
Copyright year: 2020
Copyright holder: Bai et al.
License: This is an open access article distributed under the terms of the Creative Commons Attribution License, which permits unrestricted use, distribution, reproduction and adaptation in any medium and for any purpose provided that it is properly attributed. For attribution, the original author(s), title, publication source (PeerJ) and either DOI or URL of the article must be cited.
License URL: https://creativecommons.org/licenses/by/4.0/

Keywords: OTUD5, Cervical cancer, Bioinformatics, Co-expression genes, Protein–protein interaction

Funding: National Natural Science Foundation of China 81702743 China Postdoctoral Science Foundation 2018M640612, 2019T120568 National Natural Science Foundation of China 81702743 This project is supported by grants from the National Natural Science Foundation of China [81702743] and the China Postdoctoral Science Foundation [2018M640612, 2019T120568] and the National Natural Science Foundation of China [81702743]. The funders had no role in study design, data collection and analysis, decision to publish, or preparation of the manuscript.

==============================
OTU deubiquitinase 5 (OTUD5), as a member of the ovarian tumor protease (OTU) family, was previously reported to play important roles in DNA repair and immunity. However, little is known about its function in tumors. Cervical cancer is a malignant tumor that seriously endangers the lives of women. Here, we found that low expression of OTUD5 in cervical cancer is associated with poor prognosis. Its expression is associated with tumor stage, metastatic nodes and tumor subtypes such as those related to the phosphatidylinositol–3–kinase (PI3K)–AKT signaling, epithelial-mesenchymal transition (EMT) and hormones. In addtion, we analyzed the coexpressed genes, related miRNAs, transcription factors, kinases, E3s and interacting proteins of OTUD5. We demonstrated that OTUD5 affects the expression levels of WD repeat domain 45 (WDR45), ubiquitin-specific peptidase 11 (USP11), GRIP1 associated protein 1 (GRIPAP1) and RNA binding motif protein 10 (RBM10). Moreover, hsa-mir-137, hsa-mir-1913, hsa-mir-937, hsa-mir-607, hsa-mir-3149 and hsa-mir-144 may inhibit the expression of OTUD5. Furthermore, we performed enrichment analysis of 22 coexpressed genes, 33 related miRNAs and 30 interacting proteins. In addition to ubiquitination and immunology related processes, they also participate in Hippo signaling, insulin signaling, EMT, histone methylation and phosphorylation kinase binding. Our study for the first time analyzed the expression of OTUD5 in cervical cancer and its relationship with clinicopathology and provided new insights for further study of its regulatory mechanism in tumors.

Introduction

Cervical cancer is one of the most common malignancies among women worldwide and seriously affects women’s quality of life (Vu et al., 2018). Cervical cancer has a long pre-invasive phase that can be detected by clinical and histopathological examination. Human papillomavirus (HPV) has been identified as a pathogenic factor in the progression from pre-invasive to invasive cervical cancer, which is critical for the transformation of cervical epithelial cells (Goodman, 2015; Clifford & Franceschi, 2017).

Ubiquitin is a small molecule protein composed of 76 amino acids that is widely present in all eukaryotic cells, and the sequence is highly conserved. There are only three amino acids differences between yeast and humans. The full length contains seven lysine sites (K6, K11, K27, K29, K33, K48, K63).

Ubiquitination is a dynamic post-translational modification in which target protein binds to ubiquitin molecules through an enzymatic reaction. The main function of ubiquitination is to participate in the degradation of target proteins and the clearance of abnormal proteins. The synergy of three ubiquitinated enzymes is usually required: E1 ubiquitin-activating enzymes, E2 ubiquitin-conjugating enzymes, and E3 ubiquitin-ligase enzymes.

Ubiquitination, which can affect the homeostasis of the body, is an important post-translational modification (PTM) (Swatek & Komander, 2016). Ubiquitination disorders also play important roles in diseases such as cancer (Bang, Kaur & Kurokawa, 2019), neurodegenerative disease (Boland et al., 2018), osteoporosis (Fukushima et al., 2017), muscular dystrophy (Lazzari & Meroni, 2016) and immune disease (Lopez-Castejon, 2020). Ubiquitination is a reversible physiological process. Deubiquitinases (DUBs) remove ubiquitin molecules from its substrate, balancing the regulation of ubiquitin modification balanced. DUB activity can affect cell homeostasis, protein stability, and a variety of signaling pathways.

Ubiquitination is a reversible mechanism by which deubiquitinases (DUBs) remove ubiquitin from their substrates DUBs consist of six families, including the ubiquitin-specific proteases (USPs), ovarian tumor proteases (OTUs), ubiquitin C-terminal hydrolases (UCHs), Machado-Joseph disease proteins (MJDs), the Jab1/MPN/Mov (Bechara et al., 2013) metalloenzymes (JAMMs) and motif interacting with UB-containing novel DUB (MINDYs) subfamily (Nijman et al., 2005; Kwasna et al., 2018). OTU deubiquitinase 5 (OTUD5), also known as DUBA, is an important DUB in the OTU family (Mevissen et al., 2013). Substrates that have been discovered so far are: Ku80, p53, PDCD5, STP16, UBR5 and TRAF3. OTUD5 plays an important role in DNA repair and immunity. In response to genotoxic stress, OTUD5 interacts with p53-PDCD5 to reduce its ubiquitination (Luo, 2013; Park et al., 2015). OTUD5 interacts with Ku80 to positively regulate non-homologous end joining NHEJ-mediated repair (Li et al., 2019). In addition, OTUD5 stabilizes SPT16 and regulates SPT16 dependent Pol II elongation (de Vivo et al., 2019). In immunomodulation, TGF-β stimulation leads to excessive OTUD5 production. Then, OTUD5 stabilizes UBR5, causing ubiquitination of ROR γt in mobilized T cells (Rutz et al., 2015). As a negative regulator of innate immunity, OTUD5 removes the k63 chain ubiquitination of TRAF3, ultimately suppressing IFN-I (Kayagaki et al., 2007). However, the function of OTUD5 in tumors is not clear.

In the present study, we found that reduced expression of OTUD5 is correlated with poor prognosis in cervical cancer. Moreover, we also analyzed the miRNAs and transcription factors that may regulate the expression of OTUD5, as well as kinases and E3s that may mediate the posttranslational regulation of OTUD5. Finally, through the enrichment analysis of coexpressed genes, miRNAs and interacting proteins, we discovered new processes and pathways related to OTUD5.

Methods

Cells culture and transfection

HeLa cells (obtained from the American Type Culture Collection) were maintained in DMEM supplemented with 10% fetal bovine serum and 1% penicillin/streptomycin in a humidified atmosphere with 5% CO2 at 37 °C. For the knockdown assay, cells were transfected with appropriate siRNAs against OTUD5 using Lipofectamine 2000, and scrambled siRNA was used as a control. After 48 h, the cells were collected, and the efficiency of OTUD5 knockdown was verified by qRT-PCR.

RNA interference

HeLa cells were transfected with siRNAs for OTUD5 knockdown. The sequences of the siRNA against OTUD5 are as follows: #1: 5′-GGGCUGGGCCUGCCAU CAUUC-3′, #2: 5′-GGGCCCUCAUUCAGCAGAUGU-3′.

GEPIA

GEPIA (Interactive Analysis of Gene Expression Analysis, http://gepia.cancer-pku.cn/), (Tang et al., 2017) a web-based tool that provides fast and customizable functionality based on TCGA and GTEx data. GEPIA provides key interactive and customizable features, including gene expression, correlation analysis, and patient survival analysis. We used GEPIA to analyze the expression of OTUD5, and its relationship with transcription factors, kinases in CESC.

Oncomine analysis

DNA copy numbers of OTUD5 in CESC were determined from in the Oncomine 4.5 database. Oncomine (http://www.oncomine.org) (Rhodes et al., 2007) contains 715 gene expression data sets and data from 86,733 cancer tissues and normal tissues. The analysis of OTUD5 is based on CESC studies of Scotto Cervix. We investigated the HPV infection status of CESC patients whether HPV status correlated with the copy number of OTUD5.

UALCAN analysis

UALCAN is an interactive web portal in-depth using RNA-seq and clinical data from 31 cancer types in TCGA (Chandrashekar et al., 2017). One of the feature is that it allows analysis of the relative expression of query genes in tumors and normal samples, as well as in individual tumor subgroups, based on individual cancer stages, tumor grades, or other clinicopathological features. UALCAN is publicly available from http://ualcan.path.uab.edu. We analyzed the relationship between OTUD5 expression and clinical characteristics including CESC patient age, race, weight, lymph node metastasis, and tumor stage using UALCAN. In addition, we also analyzed the relationship between methylation and OTUD5 in CESC. In addition, the correlation OTUD5 and ELK1 in CESC was assessed via UALCAN.

LinkedOmics analysis

The LinkedOmics database (http://www.LinkedOmics.org/login.php) is a web-based platform for analyzing 32 cancers datasets from the TCGA (Vasaikar et al., 2018). LinkedOmics’ LinkFinder module was used to study the differentially expressed genes associated with OTUD5 in the TCGA CESC cohort (n = 307). Genes coexpressed with OTUD5 in CESC were statistically analyzed using Pearson correlation coefficients. All results are graphically displayed on the volcano and heat map. The LinkFinder results indicate kinase target enrichment and miRNA target enrichment associated with OTUD5. FDR <0.05 was set as the threshold.

GeneMANIA analysis

GeneMANIA (http://www.genemania.org) is a user-friendly and convenient web interface for building protein-protein interaction (PPI) networks, generating hypotheses about gene function, analyzing gene lists and prioritizing them. Functional analysis (Warde-Farley et al., 2010). The site can set the source of the network edge and has several bioinformatics methods: physical interaction with OTUD5. We visualized the physical interaction gene network using GeneMANIA.

cBioPortal analysis

The cBioPortal for Cancer Genomics (http://cbioportal.org) is an open-source tool for interactive exploration of multidimensional cancer genomics datasets and currently contains 308 cancer studies (Gao et al., 2013; Cerami et al., 2012). We used cBioPortal to analyze OTUD5 alteration in TCGA-CESC. Search parameters included mutations and copy number alterations.

GO analysis and KEGG analysis

We performed GO analysis and KEGG analysis of coexpressed genes and interacting proteins of OTUD5 in the Metascape website (http://metascape.org). The GO and KEGG analysis of miRNA was performed with DIANA tools (http://www.microrna.gr/miRPathv3/). The threshold was set mini genes >3, P <0.05.

The Kaplan–Meier plotter

The prognostic value of the expression of miRNAs, including hsa-mir-137, hsa-mir-1913, hsa-mir-937, hsa-mir-607, hsa-mir-3149 and hsa-mir-144, was evaluated using the online database Kaplan–Meier Plotter (http://www.kmplot.com). To analyze the overall survival (OS) of patients with CESC, patient samples were split into two groups by median expression (high vs. low expression) and assessed by a Kaplan–Meier survival plot, and the results included hazard ratios (HRs) with 95% confidence intervals (CI) and log-rank p-values.

qRT-PCR

Total RNA was isolated using TRIzol reagent (Invitrogen, Carlsbad, CA). First strand cDNA was synthesized subsequently using the Fast Quant RT Kit (with gDNase) (TOYOBO, Japan) according to the product manual. Quantitative polymerase chain reaction (qPCR) was performed in three replicate wells on an ABI 7500 Real-Time PCR System (Thermo Fisher Scientific, Waltham, MA) using SuperReal PreMix Plus (SYBR Green) (Tiangen Biotech, Beijing, China). GAPDH, forward: CATGAGAAGTATGACAACAGCCT; reverse: AGTCCTTCCACGATACAAAG. WDR45, forward: GTGGTAGTAGTCCCAA GTTCTC; reverse: CGATCACGATCTTGTCATGG. TFE3: forward: AAGGAACGGCA

GAAGAAAGAC; reverse: CACTGGACTTAGGGATGAGAG. TBC1D25, forward: GCCCTTTACACAGTCCATCC; reverse: TGAAACTCAGCATCGCTCAG. RBM10, forward: TCCCAGTATTACTACAATGCTCAG; reverse: CTTCTCCTTCTTCTCTTTGC

CC. USP11, forward: CCGTGACTACAACAACTCCT; reverse: TCGTCATCTTCTTTC

TCATCCC. GRIPAP1, forward: CAGTAGCATCTCCTCCTTCAG; reverse: CTCCTCC

AGCATCCATTTCTC. The relative expression levels of OTUD5, WDR45, TFE3, TBC1D25, USP11, RBM10 and GRIPAP1 were calculated using the 2−ΔΔCT method and GAPDH was used as an internal control. Each experiment was repeated three times.

Statistical analysis

The cut-off values for OTUD5, USP11, RBM10, WDR45 and GRIPAP1 expression were determined by their median values. All analyses, including the t-test and correlation analysis were performed by GraphPad Prism. In our analysis, P values less than 0.05 were considered significant.

Results

OTUD5 is expressed at abnormally low levels in CESC with poor prognosis

To detect the role of OTUD5 in cervical cancer, we first analyzed the survival rate and OTUD5 expression in cervical cancer patients. Clinical survival data were derived from TCGA-CESC, containing 13 control samples (expression-TMP = 59.59) and 306 CESC samples (expression-TMP = 36.3). The samples were divided into a low OTUD5 expression group (n = 98) and a high OTUD5 expression group (n = 193) according to the median (Fig. 1A). Kaplan–Meier survival analysis revealed that low mRNA expression of OTUD5 was significantly correlated with poor overall survival (P = 0.0031). Then, we evaluated the expression level of OTUD5 in the TCGA database (Fig. S1A). The expression of OTUD5 in cervical cancer tissues was lower than that in normal tissues (Fig. 1B). In addition, OTUD5 was significantly underexpressed in cervical cancer cell lines(HT-3 and DoTc2-4510) according to COSMIC database analysis (Fig. S1B). To explore the underlying mechanisms, we first analyzed mutations in OTUD5 in CESC. The missense mutation frequency of OTUD5 was approximately 1.57% (Fig. 1C). Mutations included P478Q (in endocervical adenocarcinoma) and Q199R and E241K (in cervical squamous cell carcinoma). However, the current reports demonstrated that only mutations in the S177A and C224S sites affect the deubiquitinase activity of OTUD5 (Li et al., 2019; Huang et al., 2012; Wu et al., 2020). For the copy number variation, deep deletion samples (n = 2) and shallow deletion samples (n = 42) exist in CESC patients (Fig. 1D). Additionally, methylation affected OTUD5 expression in CESC (Fig. 1E). There was no meaningful difference in methylation levels between the tumor and normal groups (Fig. 1F). OTUD5 copy number, mRNA and mutations all may affect its expression in cervical cancer, and regulate the progression of tumors. Taken together, the low expression of OTUD5 is associated with poor prognosis in cervical cancer and OTUD5 is a potential tumor suppressor gene. In addition, OTUD5 undergoes mutational inactivation and loss of copy number in cervical cancer tissues. Under this double attack, the ability OTUD5 to inhibit cervical cancer is suppressed.

Figure 1 The expression of OTUD5 in CESC.

(A) Overall survival of CESC patients (TCGA). (B) Box plot showing the transcripts per million (TPM) of OTUD5 in CESC (TCGA) from the GEPIA database. (C-D) Scatter plot showing mutation and copy number alteration of OTUD5 in CESC patients using cBioPortal. (E) The correlation of OTUD5 and methylation in CESC data from MethHC (Data S1). (F) Box plot showing OTUD5 methylation in CESC via UALCAN. The beta value indicates the level of DNA methylation ranging from 0 (unmethylated) to 1 (fully methylated).

The association of OTUD5 expression with clinicopathologic characteristics in cervical cancer

Next, we analyzed the relationship between OTUD5 and clinicopathological aspects. OTUD5 was negatively related to tumor stage and number of lymph nodes (Figs. 2A–2B). The expression levels of OTUD5 showed no difference in groups classified by age, race, or weight (Fig. 2C–2E). Since high-risk HPV infection is a key step in cervical carcinoma, we next analyzed the patients infected with HPV in the Oncomine-Scotto Cervix (Scotto et al., 2008). We found that the copy number of OTUD5 was not different in patients with or without HPV infection (Fig. 2F). We speculated that OTUD5 may be regulated after transcription, for example, by miRNA and PTM. Next, we analyzed the relationship of OTUD5 in subtypes associated with EMT, hormones and PI3K-AKT. Epithelial-mesenchymal transition (EMT) is one of the main causes of cancer metastasis (Qureshi & Rizvi, 2015). EMT-activated transcription factors are invasive and motility characteristics for cervical cancer (Jiang et al., 2017). The expression level of OTUD5 was lower in EMT related subtypes (Fig. 2G). Hormones are not directly involved in metabolism, but can promote or inhibit the body’s original metabolic processes by regulating certain proteins. For example, oral contraceptives can affect the secretion of hormones and increase the risk of cervical cancer (Smith et al., 2003). OTUD5 is also expressed at a lower level in hormone-related subtypes (Fig. 2H). In cervical cancer, activation of PI3K-AKT signaling is closely related to tumor progression and poor prognosis (Jiang et al., 2017). Consistently, OTUD5 is also downregulated in PI3K-AKT related subtypes (Fig. 2I).

Figure 2 The relationship between OTUD5 expression and clinical pathology of cervical cancer.

(A) Boxplot showing relative expression of OTUD5 in tumor stage (stage1, stage2, stage3, stage4 or stage5) from CESC patients via UALCAN. (B) Boxplot showing relative expression of OTUD5 in nodal metastasis status (Normal, N0 or N1) from normal and CESC patients via UALCAN. (C–E) Boxplots showing relative expression of OTUD5 in age (aged 21–40, 41–60, 61–80, or 81–100 years), weight (Normal, Normal weight, Extreme weight, Obese or Extreme obese) and race (Normal, Caucasian, African-American or Asian) from CESC patients via UALCAN. (F) Box plot showing OTUD5 copy number in The Cancer Genome Atlas (TCGA) Scotto Cervix via ONCOMINE. (G) Boxplot showing Epithelial-Mesenchymal Transition (EMT) in CESC patients affected by OTUD5 in the GEPIA database. (H) Boxplot showing Hormone in CESC patients affected by OTUD5 in the GEPIA database. (I) Boxplot showing the PI3K-AKT the pathway in CESC patients affected by OTUD5 in the GEPIA database.

Taken together, the mRNA levels of OTUD5 in cervical tumor tissues are negatively related to tumor stage and node metastasis. OTUD5 expressed at low levels in CESC subtypes associated with EMT, hormones and PI3K-AKT.

Enrichment analysis of OTUD5 coexpressed genes in CESC

To determine the potential function OTUD5 in CESC, we analyzed genes associated with OTUD5 in the LinkedOmics database. The RNA-seq data were obtained from 307 samples from TCGA-CESC. In the volcano plot, 3907 genes (dark red dots) displayed meaningful positive correlations with OTUD5, whereas 3,660 genes (dark green dots) displayed meaningful negative correlations (Fig. 3A). The 50 significant gene sets positively related to OTUD5 are shown in the heat map (Fig. 3B). Among them, we selected 22 genes had have lower expression correlated with poor overall survival in CESC for further analysis (Table 1). The GO term analysis performed by using the Metascape database is shown in Figs. 3C and 3D. These genes were mainly involved in modification-dependent macromolecule catabolic processes (GO:0019941), modification—dependent protein catabolic processes (GO:0043632), ubiquitin—dependent protein catabolic processes (GO:0006511), proteolysis involved in cellular protein catabolic processes (GO:0051603), cellular protein catabolic process (GO:0044257) and cellular component disassembly (GO:0022411) (Fig. 3C). The top 3 noteworthy terms of the GO functional analysis 561416901were561416901AAdministrator5614169011736141691 modification-dependent protein binding (GO:0140030), methylated histone binding (GO:0035064) and histone binding (GO:0042393) (Fig. 3D).

Figure 3 GO functional annotation and pathway enrichment of OTUD5 Co-expressed genes in TCGA-CESC.

(A) A Pearson test was used to analyze correlations between OTUD5, and genes differentially expressed in CESC. (B) Heat maps showing genes positively correlated with OTUD5 in CESC (TOP 50). (C–D) Enrichment analysis of the genes altered in the OTUD5 co-expression genes in cervical cancer. The box plot displays the enrichment results of the top 22 genes altered in the OTUD5 co-expression genes in CESC. (C) Biological processes. (D) Molecular functions.

Next, we analyzed the correlation between these coexpressed genes with OTUD5 in GEPIA (Table 1). Four genes USP11 (Pearson correlation = 0.72, P = 2.38E−23), RBM10 (Pearson correlation = 0.73, P = 5.27E−43), WDR45 (Pearson correlation = 0.76, P = 1.63E−57) and GRIPAP1 (Pearson correlation = 0.84, P = 2.09E−45), had strong positive correlations with OTUD5 (Figs. 4A–4D). USP11 is a member of the USP family protein that is involved in several cancers including cervical (Lin, Chang & Yu, 2008), lung, liver, brain and colon cancer. RBM10 is a member of the RNA-binding protein (RBP) family and is known for its role in mRNA splicing. Preliminary works have shown that RBM10 can increase apoptosis and inhibit cell proliferation (Zhao et al., 2017; Bechara et al., 2013). WDR45, which encodes a WD40 repeat-containing PtdIns(3)P binding protein, is crucial for the basic autophagy pathway (Zhao et al., 2015). GRIPAP1 (GRIP-related protein 1) is a neuron-specific endosome protein that is identified as part of the AMPAR complex, as it directly interacts with AMPAR. GRIPAP1 has been reported to play an important role in promoting the maturation of recovered endosomes (Hoogenraad et al., 2010). In addition, the low expression of these genes in CESC is related to poor prognosis in CESC (Figs. 4E–4H). To clarify the relationship of these genes with OTUD5, we knocked down OTUD5 in HeLa cells and then performed qRT-PCR to detect their mRNA levels. We found that USP11, RBM10, WDR45 and GRIPAP1 were downregulated after knocking down OTUD5 (Figs. 4I–4L). OTUD5 and its coexpressed genes may regulate the ubiquitination of proteins and affect the occurrence and development of cervical cancer.

Table 1 Co-expression with OTUD5 in CESC from LinkedOmics.

Genes	Gene expression log2 (TPM + 1) for log-scale.	Prognose	Correlation	
	Normal	Tumor			
GPKOW	4.2	4.5	Favorable	0.79	
CCDC22	3.8	2.4	Favorable	0.76	
WDR45#	6.3	5.6	Favorable	0.65	
WDR13#	7.1	6.5	Favorable	0.38	
ARAF#	5.7	5.6	Favorable	0.48	
GRIPAP1#	5.3	5.0	Favorable	0.84	
UXT	7.1	7.3	Favorable	0.54	
RBM10#	5.2	5.2	Favorable	0.73	
TSR2	5.5	5,5	Unfavorable	0.73	
TIMM17B	5.4	6.2	Favorable	0.57	
PQBP1#	6.5	6.4	Favorable	0.51	
NDUFB11	6.0	7.1	Favorable	0.40	
HSD17B10	6.2	7.0	Favorable	0.51	
TBC1D25#	3.4	3.1	Favorable	0.70	
SUV39H1	2.2	3.7	Favorable	0.67	
FTSJ1	4.9	5.7	Favorable	0.50	
PHKA2#	4.7	3.8	Favorable	0.65	
SPIN2B#	3.7	2.9	N/A	0.51	
USP11#	6.4	5.0	Favorable	0.72	
IGBP1#	6.0	5.4	N/A	0.50	
APEX2	3.3	4.5	Favorable	0.58	
LAS1L	5.1	5.0	Favorable	0.51	
RNF113A	3.6	4.0	Favorable	0.45	
FAM104B	4.1	4.3	Favorable	0.45	
TFE3	5.6	4.6	Favorable	0.70	
TSPYL2	7.5	3.7	Unfavorable	0.63	
ELK1	4.0	4.5	N/A	0.71	
PRAF2	5.5	4.5	Favorable	0.48	
IDH3G	5.9	5.9	Favorable	0.41	
MAGED2#	7.5	6.2	N/A	0.60	
HDAC6#	6.4	5.1	Favorable	0.58	
PDHA1	6.6	7.0	Favorable	0.49	
CDK16	5.7	6.4	Favorable	0.47	
LOC550643	4.8	5.0	N/A		
SLC25A14#	3.8	3.6	Favorable	0.55	
PIN4	4.2	4.7	N/A	0.39	
EBP	4.6	7.3	Favorable	0.36	
LOC401588	2.5	1.4	Favorable	0.59	
USP27X#	2.0	1.6	Favorable	0.66	
SPSB3#	6.2	5.4	Favorable	0.42	
Cxorf40A	4.0	4.2	Favorable	0.41	
C17orf59	2.8	2.6	Favorable	0.35	
SLC35A2	4.0	4.8	N/A	0.63	
FAM156A#	4.6	3.3	Favorable	0.53	
GABARAP#	9.7	9.0	Favorable	0.39	
NKAP	2.8	3.0	Favorable	0.49	
RBMX2#	4.7	4.5	Favorable	0.43	
UBA1	7.1	7.3	N/A	0.60	
MSL3#	4.7	4.4	Favorable	0.63	

The miRNAs and transcription factors that may regulate the mRNA expression of OTUD5 in CESC

To identify the upstream molecules that regulate the mRNA levels of OTUD5 in cervical cancer, we analyzed miRNAs and transcription factors related to OTUD5. We first evaluated the miRNA-OTUD5 network in CESC using Linked Omics. In the volcano plot, 475 miRNAs (dark red dots) displayed meaningful positive correlations with OTUD5, and 334 miRNAs (dark green dots) displayed significant negative correlations (Fig. 5A). Since miRNAs usually inhibit the expression of target genes, we are more interested in negatively related miRNAs. The 33 denoting miRNAs negatively correlated with OTUD5 are shown in the heat map (Fig. 5B).

Figure 4 USP11, RBM10, WDR45 and GRIPAP1 have associated with OTUD5.

(A–D) The scatter plot shows Pearson correlation of OTUD5 expression with expression USP11, RBM10, WDR45 and GRIPAP1 in CESC patients (linkedomics). (E–H) Overall survival curve showing USP11, RBM10, WDR45 and GRIPAP1 in CESC patients (linkedomics), low expression (n = 137), high expression (n = 136). (I–L) mRNA of USP11, RBM10, WDR45 and GRIPAP1 were detected by qRT-PCR when knocked down OTUD5 in HeLa cells using siRNA.

We performed GO and KEGG analyses by searching the DIANA database (Table 2). These miRNAs were enriched in Hippo signaling (hsa04390), protein processing in the endoplasmic reticulum (hsa04141), viral carcinogenesis (hsa05203), fatty acid metabolism (hsa01212), adherens junctions (hsa04520) and proteoglycans in cancer (hsa05205) (Fig. 5C). These miRNAs were primarily involved in biological processes such as symbiosis, encompassing mutualism through parasitism (GO:0043903), cellular nitrogen compound metabolic processes (GO:0034641), cellular protein modification processes (GO:0006464), biosynthetic processes (GO:0009058), gene expression (GO:0010467) and small molecule metabolic processes (GO:0044281) (Fig. 5D). Their molecular functions were enriched in ion binding (GO:0043167), enzyme binding (GO:0019899), poly(A) RNA binding (GO:0008143), protein binding transcription factor activity (GO:0000988), RNA binding (GO:0003723), and cytoskeletal protein binding (GO:0008092) (Fig. 5E). They were predominantly localized at microtubule organizing center (GO:0005815), protein complex (GO:0031519), nucleoplasm (GO:0005654), focal adhesion (GO:0005925), cytosol (GO:0005829) and endosome (GO:0005768) (Fig. 5F). Among the top ten relevant miRNAs, we found that hsa-mir-137, hsa-mir-1913, hsa-mir-937, hsa-mir-607, hsa-mir-3149 and hsa-mir-144 were associated with poor overall survival in CESC (Figs. 5G–5H and Table 2). Therefore, these miRNAs may influence the prognosis of cervical cancer by regulating the expression of OTUD5.

Figure 5 miRNA-OTUD5 network in CESC.

(A) A Pearson test was used to analyze correlations between OTUD5, and miRNA differentially expressed in CESC. (B) Heat maps showing genes negatively correlated with OTUD5 in CESC (TOP 33). (C-F) Enrichment analysis of the miRNA negatively correlated with OTUD5 in cervical cancer. The groups display the enrichment results of the top nine miRNA that was altered in CESC. (C) pathway. (D) biological processes. (E) molecular functions. (F) cellular components. (G-L) Impact of miRNA expression on overall survival in CESC patients (Kaplan-Meier Plotter): hsa-mir-137, hsa-mir-1913, hsa-mir-937, hsa-mir-607, hsa-mir-3149 and hsa-mir-144.

Table 2 Overall survival of miRNA that target OTUD5 in CESC.

miRNA	Low expression	High expression	Prognose	P value	HR	
Has-mir-584	n = 22	n = 185	N/A	0.1900	1.39 (0.85–2.29)	
Has-mir-137	n = 156	n = 151	Unfavorable	0.0280	1.68 (1.05–2.69)	
Has-mir-1293	n = 83	n = 224	N/A	0.1800	0.71 (0.43–1.18	
Has-mir-1913	n = 227	n = 80	Unfavorable	0.0210	1.78 (1.09–2.93)	
Has-mir-937	n = 129	n = 178	Unfavorable	0.0023	2.28 (1.32–3.94)	
Has-mir-9-3	n = 76	n = 231	Favorable	0.0310	0.58 (0.35–0.96)	
Has-mir-607	n = 209	n = 98	Unfavorable	0.0089	1.86 (1.16–2.99)	
Has-mir-3149	n = 228	n = 79	Unfavorable	0.0054	2.00(1.22–3.29)	
Has-mir-144	n = 81	n = 226	Unfavorable	0.0140	2.09 (1.14–3.81)	
Has-mir-103-1	n = 159	n = 148	N/A	0.0420	1.62 (1.01–2.61)	

The transcription factors that regulate OTUD5 were previously unknown. Next, we predicted the transcription factors of OTUD5 using QIAGEN, ConTra v3 and PROMO. As shown in Table 3, there were 4 potential candidates from QIAGEN, 7 potential candidates from ConTra v3 and 9 potential candidates from PROMO. We also examined the correlation between these transcription factors and OTUD5. The expression level of OTUD5 was significantiy correlated with the transcription factors SP1, SP2, SP3, SP4, MAZ, MAZR, ELF4, IRF-2, YY1, TFIID, FOXP3, GTF2I, NF-1, ELK1 and P53 in CESC (Table 3). Therefore, we speculated that transcription factors may affect the expression of OTUD5 in cervical cancer.

Table 3 Transcription factor of OTUD5 in QIAGEN, ConTra v3 and PROMO.

Database	Gene	Correlation	P value	Database	Gene	Correlation	P value	
QIAGEN	EST1	0.12	0.038	PROMO	IRF-2#	0.33	2.7e–09	
	NKX2-5	−0.07	0.250		RXRA	0.10	0.084	
	PAX4	−0.02	0.680		YY1#	0.21	0.001	
	PPARG	0.07	0.250		XBP1	0.13	0.021	
ConTra v3	SP1#	0.34	6.2e–10		Cebpb	−0.04	0.480	
	SP2#	0.40	7.1e–13		PAX5	0.02	0.720	
	SP3#	0.26	6.1e–06		TFIID#	0.21	0.001	
	SP4#	0.32	7.6e–09		FOXP3#	0.27	1.9e–06	
	MAZ#	0.23	4.0e–05		GR	0.12	0.039	
	MAZR#	0.37	1.9e–11		GTF2I#	0.31	4.7e–08	
	KLF4	0.02	0.760		NF-1#	0.32	9.1e–09	
	KLF5	0.06	0.300		ESR1	0.13	0.019	
	KLF16	−0.02	0.720		SLEB11	−0.02	0.780	
	TCN1	−0.11	0.058		ETS1	0.12	0.038	
	ELF4#	0.28	4.2e–07		ELK1#	0.71	2.2e–47	
					TFAP2A	0.10	0.100	
					SP1#	0.34	0.038	
					HMGA1	0.10	0.130	
					P53#	0.37	2.6e–11	

Protein–protein interaction network (PPI) and enrichment analysis of proteins interacting with OTUD5

As a DUB, OTUD5 always functions by binding to another protein and deubiquitinating it. Through proteome analysis, we found 30 interacting proteins of OTUD5 and displayed them in a physical interaction network using GeneMANIA (Fig. 6A). The five most reliable proteins were ARPC3, LONRF2, GPX4, LANCL2 and GYS1 (Fig. 6A).

Figure 6 Functional annotations of OTUD5 interaction proteins.

(A) Protein-protein interaction network of OTUD5 (GeneMANIA). (B–D) Enrichment analysis of the protein interacted with OTUD5. The groups display the enrichment results of the top 30 interaction genes with OTUD5 in CESC. (B) Molecular functions. (C) Biological processes. (D) Cellular components.

Next, we performed GO enrichment analysis of these 30 interacting proteins. Since OTUD5 is a DUB, it is not surprising that the interacting proteins are enriched in ubiquitin protein ligase binding (GO:0031625) and ubiquitin-like protein ligase binding (GO:0044389) (Fig. 6B). In addition, these proteins also have the function of binding GTPases (GO:0051020) and kinases (GO:0019900) (Fig. 6B). Moreover, we noticed significant enrichment of functions corrected with actin binding (GO:0003779) and actin filament binding (GO:0051015) and processes related to positive regulation of organelle organization (GO:0006996), actin filament—based process (GO:0030029) and actin cytoskeleton organization (GO:0030036) (Fig. 6B and 6C). This suggests that OTUD5 may affect the formation of the cytoskeleton by interacting with these proteins. OTUD5 has been found to interact with UBR5 and TRAF3 and regulate immune reactions (Rutz et al., 2015; Kayagaki et al., 2007). Consistent with this, these proteins are primarily involved in the regulation of immune response—activating signal transduction (GO:0002757), activation of immune response (GO:0002253), and immune response—regulating signaling pathway (GO:0002764) (Fig. 6C). These proteins are mainly localized to the cell–cell junction (GO:0005911), cell cortex (GO:0005938), focal adhesions (GO:0005925), cell-substrate adheres junctions (GO:0005924), cell-substrate junctions (GO:0030055) and cytoplasmic regions (GO:0099568) (Fig. 6D). The numerous components of the cytoskeleton, actin and tubulin, are highly integrated and their functions are well coordinated in normal cells. In contrast, mutations and abnormal expression of cytoskeletal and cytoskeleton-related proteins play an important role in the ability of cancer cells to resist chemotherapy and metastasis (Fife & Kavallaris, 2014; Hall, 2009). OTUD5 may regulate the occurrence or metastasis of cervical cancer through its interacting protein. Here, these analyses revealed that the localization of the above proteins on the cell membrane and may affect the information transfer between cells by regulating cell signaling, such as apoptosis, WNT/β-catenin, PI3K-AKT, p53, NF-κB signaling pathways and EMT. Signaling pathways affect the occurrence and development of multiple tumors. KEGG analysis illustrated enrichment in the insulin signaling pathway (hsa04910), indicated by enrichment of genes such as GRB2, GYS1 and PKLR. Consistent with this, Insulin-like growth factor 1(IGF1R) is reported to promote cervical cancer development (Steller et al., 1996).

Phosphorylation of OTUD5 at Ser177 is critical to the enzyme activity of OTUD526.in our results, the most important of kinase-target networks related primarily to the kinases PLK1, ATM, PIK3CA, CDK5 and CHEK1. Among them, PLK1, ATM, CDK5 and CHEK1 are abnormally expressed in cervical cancer (Table 4). The predicted E3s capable of regulating OTUD5 are shown in TABLE (Table 5). The expression of NEDDL4, RNF6, MDM2 and TRIM11 is abnormal in cervical cancer. In summary, we analyzed the interacting proteins of OTUD5 and predicted the potential kinases and E3s of OTUD5.

Table 4 Kinase of OTUD5 in CESC from LinkedOmics.

Gene	Size	FDR	Gene expression log2 (TPM + 1) for log-scale	
			Normal	Tumor	
PLK1	38	0.0075271	1.6	5.3	
ATM	50	0.015054	4.3	2.7	
PIK3CA	4	0.067242	3.2	3.2	
CDK5	28	0.074669	3	4.2	
CHEK1	46	0.078282	1.4	4.2	

Table 5 Ubiquitin ligase of OTUD5 in CESC from UbiBrowser.

Genes	SCORE	Gene expression log2(TPM + 1) for log-scale.	Genes	SCORE	Gene expression log2(TPM + 1) for log-scale.	
		Normal	Tumor			Normal	Tumor	
RNF180	0.719	1.9	0.1	FBXO3	0.623	4.6	4.0	
NEDD4L#	0.705	3.0	4.7	HERC2	0.623	5.1	4.8	
RNF6#	0.694	3.4	3.9	TRIM11#	0.610	3.8	4.6	
MARCH1	0.694	1.8	1.3	KLHL13	0.610	4.1	1.9	
MDM2#	0.668	5.0	6.0	MARCH8	0.610	3.6	2.8	
SYVN1	0.667	5.3	4.8	RFPL4A	0.610	–	–	
PIAS4	0.648	3.7	3.8	MARCH7	0.608	5.2	5.3	
MIB1	0.642	3.4	2.8	RNF216	0.608	4.1	3.4	
ZC3HC1	0.623	3.7	3.6	MIB2	0.594	6.1	5.1	
				UBE3C	0.594	4.6	4.6	

Discussion

Increasing epidemiological data have shown that cervical cancer is the most popular malignancy among women worldwide. The long-term survival rate of CESC patients is low due to the ease of recurrence and metastasis. The occurrence and development of malignant tumors is usually a multistage complex process, involving a variety of genes and signal transduction pathways. Many genes, including BAX and p53 associated with Bcl-2, have been reported to play an important role in the progression of cervical cancer (Karlidag et al., 2007). Therefore, the study of abnormal CESC gene changes is vital to further clarify the mechanisms of occurrence and development of cervical cancer.

OTUD5 was previously reported to play important roles in DNA damage and immune regulation. Little is known about its function in tumors. In the present study, bioinformatic analysis via high throughput RNA-sequencing data from TCGA indicated that low OTUD5 expression in CESC and was correlated with leading clinical features (tumor stage, nodal metastasis status) and overall survival. This revealed that OTUD5 may act as a possible biomarker of prognosis and treatment target in CESC.

Phosphorylation of Thr308 and Ser473 leads to the activation of PI3K-AKT signaling (Jiang et al., 2017). OTUD5 was reported to affect the phosphorylation of Akt in cervical tumor cells (Yin et al., 2019). In this analysis, the expression level of OTUD5 was consistently downregulated in PI3K-AKT subtype tumors. The proteins interacting with OTUD5 were involved in the function of kinase binding.

In recent years, there have been increasing number of reports on the relationship between epigenetic modification and tumors (Wu et al., 2020). The methylation of histones is closely related to tumors (Audia & Campbell, 2016). Our analysis suggested that although OTUD5 is methylated in cervical cancer, its coexpressed genes may affect the progression of cervical cancer by regulating methylation.

In addition, HPV, as a virus, has been identified as a pathogenic factor in cervical cancer. The miRNAs related to OTUD5 are enriched in viral carcinogenesis. They may regulate cervical cancer caused by HPV infection by regulating OTUD5. The Hippo signaling pathway is reported to be active in cervical cancer (Zhao et al., 2017). The functions of OTUD5 in the Hippo pathway are not clear. According to our results, miRNAs associated with OTUD5 are enriched in Hippo pathway. The new regulatory relationship between miRNA and OTUD5 may be involved in the regulatory network of the Hippo pathway in cervical cancer.

In cervical cancer, the predicted kinases have been reported to perform different functions. For example, the c-ABL-PLK1 axis is a new prognostic marker and treatment target in cervical cancers (Yang et al., 2017). Osthole treatment also sensitized cervical cancer to irradiation, revealing promoted DNA damage and prohibited ATM/NF-κB signaling (Che et al., 2018). CDK5 phosphorylates p53 at serine 20 and serine 46 residues, thereby promoting its recruitment to p21 and BAX promoters in cervical cancer (Ajay et al., 2010). CHEK1 and p-CHEK1 contribute to the development of cervical cancer (Indra et al., 2011). This illustrates a new model between phosphorylation and ubiquitination.

Conclusions

In summary, through bioinformatics analysis, we obtained a comprehensive view of the role of OTUD5 in cervical cancer. OTUD5 is a protein expressed at low levels and associated with poor prognosis. Analysis of OTUD5-related genes, miRNAs and transcription factors; proteins interacting with OTUD5; and kinase and E3s targeting OTUD5 and further analysis of GO and pathway enrichment and PPIs, may provide new regulatory mechanisms associated with OTUD5 in cervical cancer.

Supplemental Information

Supplemental Information 1 The expression of OTUD5 and p53

(A) Expression of OTUD5 in 31 types of normal and tumor samples. Transcripts per million (TPM) of OTUD5 in CESC (TCGA) from the GEPIA database. (B) Expression of OTUD5 in cervical cancer cell lines (HT-3 and DoTc2-4510) through COSMIC database analysis. (C) Box plot showing the transcripts per million (TPM) of p53 in CESC (TCGA) from the GEPIA database.

Click here for additional data file.

Supplemental Information 2 qPCR

mRNA levels of USP11, RBM10, WDR45 and GRIPAP1 were detected by qRT-PCR when OTUD5 was knocked down in HeLa cells using siRNA.

Click here for additional data file.

Supplemental Information 3 Copy number alteration

Scatter plot showing mutation and copy number alteration of OTUD5 in CESC patients using cBioPortal.

Click here for additional data file.

Supplemental Information 4 HPV infection in Scotto Cervix

Box plot showing OTUD5 copy number in the TCGA Scotto Cervix dataset via ONCOMINE.

Click here for additional data file.

Supplemental Information 5 Cell lines

Expression of OTUD5 in cervical cancer cell lines (HT-3 and DoTc2-4510) through COSMIC database analysis.

Click here for additional data file.

Supplemental Information 6 Genes positively correlated with OTUD5 in CESC

Click here for additional data file.

Supplemental Information 7 Methylation

The correlation of OTUD5 and methylation in CESC data from MethHC (http://methhc.mbc.nctu.edu.tw/php/index.php).

Click here for additional data file.

Supplemental Information 8 Overall survival of CESC patients (TCGA)

Click here for additional data file.

Supplemental Information 9 The correlation of OTUD5 with WDR45-GRIPAP1-USP11-RBM10

The scatter plot shows the Pearson correlation of OTUD5 expression with USP11, RBM10, WDR45 and GRIPAP1 expression in CESC patients (LinkedOmics).

Click here for additional data file.

Supplemental Information 10 Overall survival curve showing USP11, RBM10, WDR45 and GRIPAP1 in CESC patients (linkedomics), low expression (n=137), and high expression (n=136)

Click here for additional data file.

Supplemental Information 11 OTUD5 and methylation in CESC data

Click here for additional data file.

Additional Information and Declarations

Competing Interests

Author Contributions

Data Availability

The authors declare there are no competing interests.

Mixue Bai conceived and designed the experiments, analyzed the data, prepared figures and/or tables, and approved the final draft.

Yingying Che performed the experiments, analyzed the data, authored or reviewed drafts of the paper, and approved the final draft.

Kun Lu performed the experiments, analyzed the data, prepared figures and/or tables, and approved the final draft.

Lin Fu conceived and designed the experiments, analyzed the data, authored or reviewed drafts of the paper, and approved the final draft.

The following information was supplied regarding data availability:

The raw measurements are available in the Supplemental Files.

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
