# Peer review of "Analysis of deubiquitinase OTUD5 as a biomarker and therapeutic target for cervical cancer by bioinformatic analysis"

_PeerJ, doi:10.7717/peerj.9146_

## Round 0.1 · original submission · Major Revisions

Three experts in the field evaluated this submission. They all have concerns related to several parts of this work. They also request an English revision of the text. In my view, this paper needs a major revision.

Reviewer 1 ·

Basic reporting

The article needs linguistic improvement

Experimental design

No comment. Seems scientifically sound

Validity of the findings

I would like to suggest authors to prove the bioinformatic data with experimental works, like protein level expression of OTUD5.

Reviewer 2 ·

Basic reporting

The English language should be improved to ensure that an international audience can clearly understand the text. It gets extremely difficult in the 'Discussion' section to understand what authors are trying to say. The language of all the sections needs to be improved.

Lot of acronyms have been used throughout the manuscript, even in the abstract (e.g. OTU, EMT, WDR-45 etc.) , without their full form. Please provide full form of all the acronyms used.

Literature references should be provided in the introduction and discussion section (e.g. Line 50, 302 etc.)

Experimental design

In the present study, using bioinformatic analysis, the authors have shown that lower expression of OTUD5 is associated with poor prognosis of Cervical Cancer and it's expression is associated with tumor stage, metastatic nodes and tumor subtypes.

In the results section 3.1, data analysis from TCGA database shows that there are missense mutations and copy number variations and deletions in OTUD5 gene in Cervical Cancer patients. Authors need to comment on the the upstream and downstream regulatory pathways which might be responsible for their observation.

In the results section 3.2, authors need to comment on why there is no difference in OTUD5 expression in HPV infected patients.

In the results section 3.3, authors have studied genes, which are correlated with OTUD5 expression in Cervical Cancer. Four highly positively correlated genes: USP11, RBM10, WDR45 and GRIPAP1 were chosen for further analysis.To clarify, their relationship with OTUD5, authors have knocked down OTUD5 in HeLa cells and showed that they are positively correlated. However, they need to comment on the fact that how this correlation affects poor prognosis of CESC? What happens to OTUD5 expression if USP11, RBM10, WDR45, GRPAP1 is knocked down? From this data, can we draw any conclusion on how these genes are related to each other and lead to poor prognosis?

In Results section 3.4 and 3.5, authors have described the miRNAs and transcription factors that regulate mRNA expression of OTUD5 and proteins that interact with OTUD5. They describe their observation but need to discuss how these regulation of OTUD5 expression and protein-protein interaction might cause poor prognosis of CESC.

Validity of the findings

I thank authors for providing all the underlying data and statistical analysis. However, they need to comment on their observation. Only stating the data is not sufficient.

Reviewer 3 ·

Basic reporting

Significant efforts are needed to improve the grammar in the manuscript as well as the general use of English (some sentences do not make sense - perhaps an error in translation?). Please also consider one of the software available to check and correct grammar. Additionally, it is highly recommended that authors find a colleague or a collaborator with good writing skills in English and have the manuscript reviewed. There are several instances where sentences that do not make any sense logically or scientifically.
- The first word in the abstract, abbreviation OTU needs to be defined.
- First sentence in abstract:
"OTU deubiquitinase 5 (OTUD5), as a member of the OTU family, was previously reported play important roles in DNA repair and immunity." 'to' is missing before 'play'.
Define abbreviations: PI3K-AKT, EMT
Add some information and/or descriptive terms about the genes being discussed: WDR40, USP11, GRIPAP1, and RBM1 to improve scientific relevance.
Avoid repetition of words such as 'Furthermore' in consecutive sentences
The title of the manuscript needs to be changed too, in its current format it is not scientifically accurate. Consider something along the lines of
"Bioinformatics analysis reveals deubiquitinase OTUD5... "
Introductions are to be used as an opportunity to clearly set background for the research done or the results being presented. In several instances author go into expressing opinions or use sentences that read more like a discussion than an introduction. Also mostly these are unnecessary interpretations/expression of opinions. Just state facts and cite relevant literature.
Example:
Line 40-42:
Although it has entered the era of targeted therapy and immunotherapy, traditional surgery, radiotherapy, and chemotherapy are still the most common treatments for cervical cancer.

This sentence is unnecessary in the introduction. Neither the treatment options or the trends in the field are a focus of this manuscript.


Check grammar:
43-44

Authors need to rewrite the introduction. Very little to no information is provided about the cellular role of otulins and ubiquitination.
Include in the introduction:
Otulins
Ubiquitination machinery
he roles associated with ubiquitination status of the cellular proteins
Background information on OTU5
known interactions of otulins, particularly OTU5

Reread the manuscript and update citations. Appropriate citations are missing in several instances.

Experimental design

There are serious flaws in the experimental design:
- No efforts have been done, or authors have not mentioned, whether the databases they are using have the curated information that would be appropriate to make scientific conclusions.
- The datasets the authors have used as control is significantly smaller than the CESC samples, invalidating several of the results based on these datasets.
Figure 1B: Not a meaningful comparison if taken into account the number of samples in control vs CESC.
Also due to such a high variability in transcript values in CESC, it cannot be conclusively interpreted that the expression of OTUD5 in cervical cancer tissues was lower than that in normal tissue. A subset of samples in CESC has the exact identical distribution to that of normal samples.

- Figure 1A:
It is unclear what data the authors are plotting and interpreting in Figure 1A. 'Survival of OTUD5' is not a scientifically sound observation or a measurement. Explain the rationale, observation, and the plot further.
Figure 2A:
Again, high variability in the number of samples in each dataset.
In Stage 1 and Stage 2 samples, n = 161 and 69, respectively, the transcript level of OTUD5 range from 20 to 80, this is contracted against the 'normal', n=3. This is not a valid comparison (1) due to the high variability of transcript levels and (2) high variability of the number of samples in the datasets being compared. This weakens, if not invalidates, the interpretation of data.
Authors need to include some control that is known to maintain stable expression and this can be used for meaningful comparison.

Figure 3A:
Levels of other proteins are compared against that of OTUD5 to identify the genes whose expression is positively or negatively correlated with that of OTUD5. But, Figure 1 shows that OTUD5 expression is highly variable in CESC samples. In such a case, comparing the gene expression profile for other genes to that of OTUD5 would not yield any meaningful results. All dependent results discussed in figure 3 are shaky.

Possible solution - include a housekeeping gene or a gene known to have a stable expression as a control.

Validity of the findings

Due to the flaws identified in the 'experimental design' section above, several results and conclusions are invalid without the proper controls.
Also due to the high variability of sample sizes, the comparisons are meaningless as identified above.

---

## Round 0.2 · accepted · Accept

The authors carried out all modifications indicated by the reviewers. In my view, the manuscript can be accepted as it is.

Reviewer 2 ·

Basic reporting

The English language of the manuscript has been improved from the previous version. However, it still need more work.

Please provide literature reference at line 301 in the discussion section.

Experimental design

All the comments on the previous version is addressed in the current version of the manuscript.

Validity of the findings

No comments